# Views of medical residents on a research training program: A qualitative study

**Pamela Abi Khalil[1], Gladys Honein-Abou Haidar[2], Dina El Achi[3], Lara Al-Hakim[1], Hani Tamim[1,4], Elie A. Akl[1,4]***

1 Faculty of Medicine, Clinical Research Institute, American University of Beirut, Beirut, Lebanon, 2 Hariri School of Nursing, American University of Beirut, Beirut, Lebanon, 3 Faculty of Medicine, American University of Beirut, Beirut, Lebanon, 4 Department of Internal Medicine, American University of Beirut Medical Center, Beirut, Lebanon

* ea32@aub.edu.lb

## Abstract

### Introduction

The American University of Beirut Medical Center (AUBMC) developed the Fellowship and Residency Research Program (FRRP) to provide residents and clinical fellows with a supervised and structured research experience. The objective of this study was to explore the views of FRRP participants about the training program and how to enhance it.

### Methods

*In 2018, w*e conducted a qualitative study targeting residents where we invited potential participants through email and via snowball sampling. We continued the data collection until we reached data saturation with 21 participants (N = 21) and used thematic analysis to identify emerging themes.

### Results

Thematic analysis generated four emerging themes: one related to the expectations of residents, while the remaining three themes related to how the program is working to meet those expectations, specifically: coordination, mentorship, and capacity building. For these three latter themes, we discussed the strengths and challenges related to each. In terms of coordination, few residents complained that the deadlines to submit deliverables are not evenly distributed across the years. In terms of mentorship, participants appreciated the flexibility in choosing the mentor. In terms of capacity building activities, participants found the lecture series, both real time and virtual, to be helpful. Finally, participants pointed out that the FRRP program should be mandatory.

### Discussion

To provide residents and clinical fellows with a supervised and structured research experience, we have built on our findings to introduce several changes to our program such as ensuring the availability and commitment of faculty mentors, and providing capacity building activities to the program participants.

**Data Availability Statement:** All relevant data are within the manuscript and its Supporting information files.

**Funding:** The author received no specific funding for this work.

**Competing interests:** Dr. Elie Akl directs the Clinical Research Institute, under which the program that was evaluated operates. Dr. Hani Tamim directs the Fellowship and Residency Research Program, the program that was evaluated. Ms. Abi Khalil coordinates the Fellowship and Residency Research Program, the program that was evaluated. All remaining authors have nothing to disclose. Please note that this does not alter our adherence to PLOS ONE policies on sharing data and materials.

## Introduction

Early exposure to research experience in undergraduate college, medical students and residents provides an important opportunity to foster one's academic medical career [1] and enhance research capabilities [2]. A survey of psychiatry faculty and residents in the US, reported that 98% of respondents indicated that research training should be offered to residents [3]. Research curricula instill positive attitudes toward research, particularly among residents, and thus help in achieving the use of evidence at the point of care and increased participation in international research networks [4]. In addition, scholarly productivity based on journal publication is associated with clinical performance during residency training [5]. This suggests that residents who invest substantial efforts in research are not compromised in their abilities to learn medicine and care for patients [5]. Therefore, it is essential to introduce research practices in the early academic career of medical trainees.

In 2011, the faculty of medicine at the American University of Beirut launched the Residency Research Program which is now known as the Fellowship and Residency Research Program (FRRP) [2]. The objective of the FRRP is to provide post graduate trainees with a supervised and structured research experience. The FRRP connects trainees with advisors with whom they work on conceiving, planning, and conducting a research project, and ideally publishing the findings in a peer-reviewed journal [2]. The program is mandatory and targets both categorical residents and clinical fellows. Participants are expected to carry out a research project throughout their residency or fellowship period, and ideally publish it.

Box 1 briefly describes the FRRP (its portal and roles of contributors), while Fig 1 describes the timeline of participation in the FRRP, using an example of a 3-year residency. Box 2 provides a brief description of the FRRP capacity building activities that cover the 'ABCs of research'. Please note that these details reflect enhancements introduced as a result of this study.

The program leadership recently conducted a survey of 103 residents enrolled in the FRRP to evaluate the program [2]. About a fifth (19%) of participants reported improvement in research-related tasks before and after completion of the program ($P < 0.0001$). Substantive percentages of participants reported either having published (34%) or being in the process of publishing their projects (55%). At the same time, several participants informally reported some challenges or made suggestions to improve the program. For example, 37% (SD = 18) of participants reported challenges in receiving appropriate supervision.

The objective of this study was to qualitatively explore the views of FRRP participants about the training program and how to enhance it.

## Materials and methods

### Overall design and research team

This descriptive qualitative approach was based on a focus group discussion as a method for data collection using a semi-structured interview guide (Appendix 1). The descriptive approach is a naturalistic inquiry that has a broad range of choices for theoretical or philosophical orientations, sampling techniques and data collection strategies [6]. Focus group is particularly useful to capitalize on group dynamics in order to get to a deeper and richer understanding of a phenomenon of interest. It is the collective understanding of the phenomenon that is considered. In a focus group, the investigators decentralize themselves from the discussion and play the role of moderators rather than interviewers during the data collection [7].

## Box 1. Brief description of the FRRP portal and roles of contributors

FRRP portal: An online platform where:

- Residents submit deliverables (e.g., Interest questionnaire, letter of intent, full proposal) and other program requirements (e.g., proof of required training in ethical conduct of research);

- Faculty members review submitted deliverables and provide feedback;

- Coordinator tracks the progress of the residents.

FRRP coordinator:

- Manages the FRRP portal;

- Orients residents to the program, its timeline and its deliverables;

- Follows-up with residents to ensure the timely completion of the research project (through frequent communications and in person meetings);

- Ensures residents identify their FRRP advisor/mentor;

- Ensures the advisor is aware of the progress of the resident's project;

- Organizes the clinical research open house where all graduating residents present their research work in posters and oral presentations.

Advisor:

- Meets with the resident on a regular basis to provide guidance, including on the appropriateness of the research topic;

- Reviews and approves deliverables prior to their submission on the FRRP portal. Makes sure the resident submits the deliverables on time;

- Reviews IRB application and secures its approval;

- Facilitates data acquisition (if applicable);

- Ensures successful completion of the project.

Department representative:

- Makes sure each resident identifies a faculty member as advisor and is working on a feasible scientific research project.

Department coordinator:

- Makes sure each resident identifies a faculty member as advisor;

- Follows up closely with residents to make sure they submit the deliverables before the deadlines.

The two team members who conducted data collection (PA and LA) were trained in focus group facilitation and data analysis techniques. Two senior team members (GHA and EAA) with significant experience in conducting and publishing qualitative studies provided oversight and peer observation during the focus group discussions. Following the first few

## First Year

| | |
|---|---|
| June | **Interest Questionnaire** |
| July | **CITI courses certificate** |
| August | |
| September | |
| October | **Letter of Intent** |
| November | |
| December | |
| January | |
| February | |
| March | |
| April | **Full Proposal** |
| May | |

## Second Year

| | |
|---|---|
| June | **Department presentations for methodology feedback** |
| July | |
| August | **IRB Approval** |
| September | |
| October | |
| November | |
| December | **Data Collection** |
| January | |
| February | |
| March | |
| April | |
| May | |

## Third Year

| | |
|---|---|
| June | **Data Collection** |
| July | |
| August | **Data Analysis** |
| September | |
| October | **Department presentations for results feedback** |
| November | |
| December | **Manuscript write-up** |
| January | |
| February | |
| March | **Department presentations** |
| April | **Poster creation** |
| May | **OPEN HOUSE** |

**Fig 1. Timeline for residents (in a 3-year residency) participating in the FRRP to complete their projects.**

---

### Box 2. Brief description of the FRRP capacity building activities that cover the ABCs of research

- Real time lectures: lectures by faculty members who are experts in the subject matter;

- Recorded lectures: recorded lectures by faculty members accessible on an online platform;

- Biostatistics clinic: Hands-on data analysis/statistical sessions where data analysts provide support, answer questions, and offer advice on issues relating to the statistical analyses of the FRRP projects;

- Opportunity for oral and poster presentations during an "Open house for clinical research" event that mimics national and international conferences.

discussions, debriefing sessions were held to reflect on the process of data collection and measures for improvement.

## Sampling and recruitment

We used the purposive sampling to identify potential participants followed by a snowball sampling. Our target population consisted of all residents that have completed at least one year of training in one of the post-graduate training programs at AUBMC to ensure enough exposure to the program.

The list of trainees was obtained from the FRRP database. We invited 140 by email to participate. Those who consented were approached and were asked to invite their peers to participate (snowball sampling). We kept on recruiting until we reached thematic saturation (see below). The Institutional Review Board (IRB) at the American University of Beirut has approved this study (protocol number: IM.HT.08/SBS-2017-0273).

## Data collection

We conducted the focus groups during the first half of 2018 (January to May) in private locations at AUBMC convenient to participants. Discussions were conducted in English, lasted for 50–70 minutes, and were audiotaped. The discussion started with a short description of the study and reiterated what was stated in the consent form regarding confidentiality and right to withdraw from the study.

Then, the facilitators started asking the open-ended questions with probing questions, using the semi-structured interview guide (Appendix 1). The questions focused on: responsibilities and learning experiences of residents, their views, attitudes, challenges, and suggestions. However, we allowed the participants to address other topics, hence allowing new ideas to emerge. We continued data collection until we reached data saturation (i.e., no more new information emerged).

## Data analysis

Immediately following each discussion, we transcribed the audiotaped recordings and anonymized them. We used the six steps of the thematic analysis by Braun and Clarke [8].

Step one, two team members (DA, PAK) read independently the first transcript to get acquainted with the information (familiarization). In the second step, they independently annotated line by line the transcript providing a label for each idea (coding). Then, the two team members met with a third investigator (GHA) to share, compare their independent data analysis, and resolve any disagreements. Once they reached an agreement, they continued coding the remaining transcripts independently. In step 3, the list of codes was generated and shared with the research team to discuss the relationships between codes. In step 4, a log of potential themes and sub-themes was developed. In step 5, the list of themes was further refined based on consensus reached among all research team members. As for phase 6, the findings were presented in a narrative form, and a synthesis of the results was included that summarize the views, attitudes, challenges, and suggestions of the participants. When reporting findings, we supported the narrative by quotes from the participants, and referred to the participants using codes such as FG1-P01.

## Increasing rigor

We ensured trustworthiness using several approaches. In terms of credibility, all conversations were audio recorded, transcribed verbatim, and used as the main data repository. As for

confirmability (i.e. objectivity), we used two strategies. First, as we indicated, the research team met to reach agreement on how the coding is done and thereafter met on several occasions to discuss the candidate themes. All team members were involved in the final data analysis to avoid misinterpretation of the results. Second, reflexivity. The research team was mindful to outline their personal assumptions during the analysis to avoid any subjective interpretation of findings.

We used a semi-structured interview guide to increase the dependability of our research and the decision to stop further data collection was made when saturation was reached. For authenticity, we reported disconfirming results.

## Results

### Participants

We conducted 3 focus groups with a total of 21 residents from different departments at AUBMC, including internal medicine, pediatrics, family medicine, emergency medicine, anesthesiology, and obstetrics and gynecology. Participants were within 2 years of completing their post graduating training.

The thematic analysis generated four themes: one related to the expectations of residents, while the remaining three themes related to how the program's pillars worked together to meet those expectations, specifically: coordination, mentorship, and capacity building. We discuss the strengths and challenges related to each of the latter three themes. Fig 2 provides a

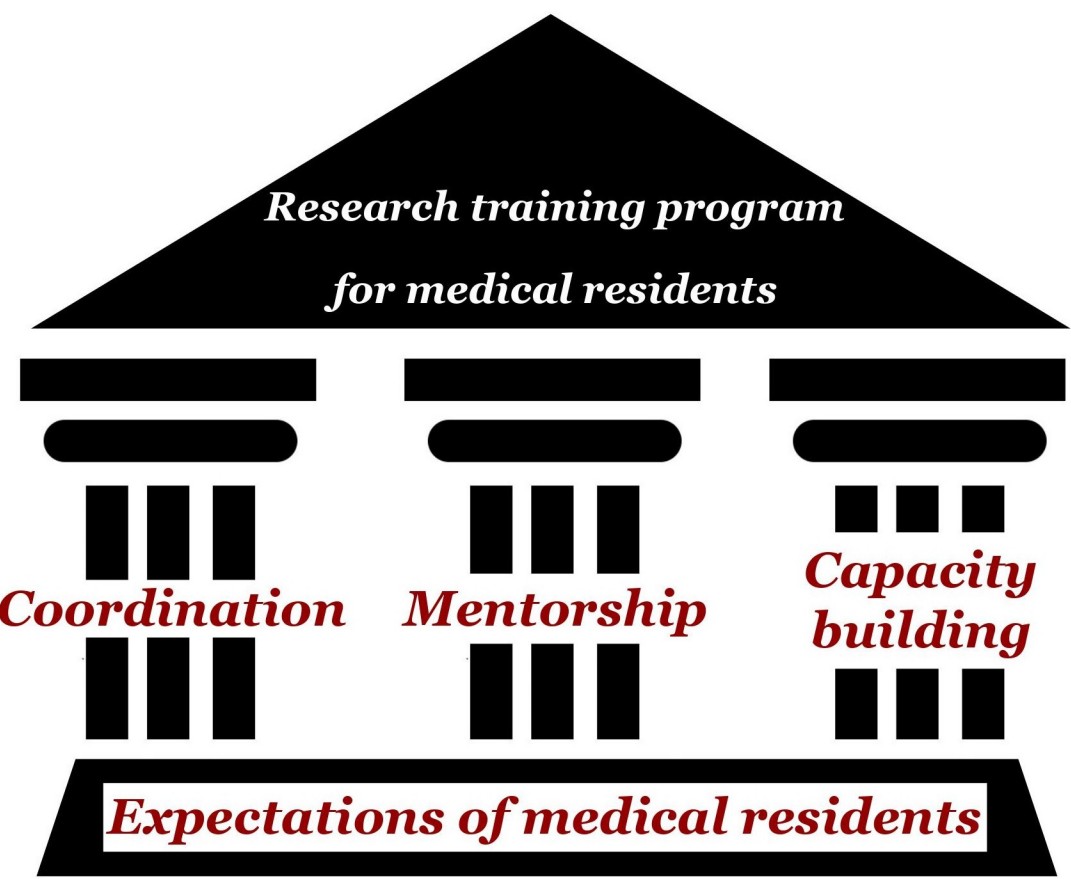

**Fig 2. Schematic representation of the four emerging themes.**

Table 1. Elements pertinent to each of the three pillars of a residency research program.

| Pillar | Elements of the pillar |
|---|---|
| Coordination | Supporting residents in:<br>• Setting a realistic timeline for the research project<br>• Submission of deliverables according to the set timeline<br>• Submission of deliverables using the online portal |
| Mentorship | • Ensuring the availability and commitment of mentors<br>• Supporting the matching of residents to mentor<br>• Encouraging residents to come up with research questions |
| Capacity building | • Educational offerings covering different aspects of research<br>• The use of interactive and hands on sessions, both real time and virtual<br>• Offering mentorship training to faculty members<br>• Organize an open house research event allowing residents to showcase their work and practice their presentation skills |

schematic representation of the four emerging themes. Table 1 describes the elements pertinent to each of the three pillars. The complete list of comments from participants pertaining to each theme can be found in the supporting information file S1 Table.

## Expectations of residents

When asked about their expectations from the program, residents described themselves as novice researchers, thus mainly required capacity building in conducting research. While one participant stated that "*some had more experience than others*" (FG1-P01), the consensus mentioned by other participants was that they lacked the "*ABCs of research*" (FG1-P02).

For example, some participants had difficulties identifying important and interesting research questions, and thus were expecting that the program will help them do so. Yet, others were expecting support for more basic skills such as statistics.

"*We were doing the data sheet and he asked me to do the table and put the values for the research, I wasn't sure, should I put them in a horizontal or vertical way, divide them by sex or just put them all together, I really did not know*"

(FG1-P01)

According to our participants there were three pillars that can set the program to be a success:

## Pillar 1: Program coordination

At its current status, the program required residents to identify a faculty mentor, a timeline for completing the research, deadlines to submit deliverables, close follow-up by the program coordinator, and the use of the portal.

The program requires the resident to identify a faculty member as a mentor, or "*as a navigator*" (FG1-P04), which encourages him/her to approach faculty members with similar research interest.

The timeline to complete the program, and depending on the type of residency, varied between three and five years. Participants highlighted the value of such an extended timeline in allowing enough time for residents to bring their research idea to maturity: "*some programs might be three, four or five years but overall enough to choose and initiate a study*" (FG1-P06).

Another benefit of having an extended timeline was the ability to change the research question as one of the participants highlighted: "*if you don't like your study you can change because you have a lot of time*" (FG1-P06). Additional strengths included the strict deadlines set for submitting the different deliverables and the close *"follow-up"* (FG1-P07) by the FRRP coordinator to complete their research on time.

Also, having the *"portal to upload our proposal. . . it's a good way"* (FG1-P04) to communicate back and forth with the program.

In terms of challenges, the participants highlighted their inability to stick to the timeline across the years and the lack of commitment of the mentor toward the research project. Although the time span to deliver products was five years, for some participants complained that the deadlines to submit the sequential deliverables were not evenly distributed across the years. For example, the bulk of the research work, as mandated by the deadlines, was to be completed during the last year of residency. Failure to deliver on time for any reason led to run into the risk of not being able to graduate.

"*It was very long in the beginning and then at the last year the deadlines become very intense. Condensed. . . And then we got stuck during the last year*"

(FG1-P09)

To address the timeline challenge, participants suggested *"making the deadlines closer"* (FG1-P09) or *"shifting everything 6 months"* (FG1-P04) to give more time to overcome barriers, if any, and still be able to graduate on time. Another suggestion was to have *"two residents work on the same research project to decrease the workload"* (FG1-P01). Participants also suggested having more coordination at the departmental level. Some suggested that the FRRP coordinators monitor progress and check on the challenges of the residents. Others suggested having in every department, a clinician scientist well versed in research as "*a reference person*" (FG1-P04), to coordinate and monitor the progress of residents.

### Pillar 2: Mentorship

Participants appreciated the flexibility in choosing the mentor. They typically based their choice of mentor on the experience of their seniors with the different attendings. Senior residents also encouraged their junior colleagues to seek mentors who have their own research assistant. One resident quoted a senior and said: "*If someone asked me what you think about choosing this mentor over someone else, if he has a research assistant I will say go for it.*" (FG1-P04). Participants highlighted, as the main strength of the program that they are "encouraged to choose" (FG1-P04) their own research question in agreement with their mentor.

In terms of challenges, participants thought there were not enough faculty members who conducted research and because there were few mentors, some "*are not available or already saturated with their mentees*" (FG1-P01) and therefore had more than they could handle. Among those in the program, they complained that they were either being busy or travelling. Thus, they were very slow, sometimes it took "*more than 6 to 7 months*" (FG2-P09) for mentors to respond to their requests, which hindered the progress of their research. Interestingly, some participants pointed to their interactions with their mentors. They said they felt intimidated by their mentors and indicated "*worrying about disappointing*" them (FG1-P04) on one hand and embarrassed to ask them questions out of fear of being "*silly questions*" (FG1-P04). So many to avoid being judged as not competent enough for the program, they would "*rather not ask*" (FG1-P01). This resulted in "*acting like you know how you are doing*" (FG2-P13), and residents

spending time figuring out things on their own. On another front, although residents had the flexibility to choose their own research questions, they ended up working on a question proposed by the mentor. However, this turned out to their advantage as they have "*shared interest*" (FG1-P04) with the attending, and likely the question would important and influential.

In terms of recommendations, more than one participant suggested that FRRP develops for each department a list of attendings that were active in research and good mentors in order to guide residents in their choices. Also, FRRP should be looking at the mentorship workload. If mentors had too many residents that were working with them, then they need to direct residents to others. One participant explained: "*Ensure that the mentor is available to the mentee*" (FG2-P12).

## Pillar 3: Capacity building

Most residents were concerned about meeting the timeline set by their mentors mainly due to their ill-preparedness in conducting research. However, when considering the specific capacity building activities, the participants found the lecture series, both real time and virtual, to be helpful. In addition, they were satisfied with the topics of the lectures as one resident said: "*the topics chosen are very good; manuscript writing, data analysis, SPSS, proposal*" (FG1-P06) They recommended"*small group*" (FG1-P08) discussions where "*it's easier to understand and residents can teach each other*" (FG1-P08) and they will be given the opportunity to share their experiences, challenges and learn from each other measures on how to overcome them. One participant shared his experience where in a small group setting his colleagues were able to guide him through a major milestone and that's what he said: "*Certain things may actually rise; several people would actually agree and it also happened with me*" (FG1-P02).

Participants also appreciated posting the lectures on the web and perceived the "clinical research open house" event as an opportunity to showcase their work and get prepared for presenting at national or international conferences. One participant mentioned that several residents present their research work orally or via a poster in international conferences and the open house was a great opportunity for them to practice since it "*mimics the conferences*" (FG1-P08).

Finally, participants pointed out that the FRRP program is very supportive and resourceful.

"*We set meetings whenever we have issues. They always find time available. Whenever the attending is there, we can ask questions, and they directly give us an answer. Last time I had a question for [mentor], she picked up the phone and called [FRRP person]. Another time someone else and she solved the issue, it was very easy.*"

(FG2-P11)

In terms of challenges, some participants thought that the capacity building sessions were "*not interactive*" (FG1-P01) enough. Also, some participants reported not taking advantage of the recorded sessions because they either forget or do not have the time. One participant states: "*But honestly we don't check them*" (FG1-P04). Other participants had time management issues.

In terms of recommendations, there was a suggestion to add an initial session to introduce research and to make the sessions mandatory for every first-year cohort. Further, participants suggested having "*a standard like first 6 months to a year where all the specialties are to attend 2 or 3 workshops*" (FG1-P04). The topics should cover all components of research.

When we asked the participants whether FRRP should be mandatory or optional, most participants thought it should be mandatory. There was further discussion on making certain

components of the program mandatory: *"We have to generalize the ABCs, like we said grand rounds for everyone, and then after a certain point this has to become personalized, individualized, so that there is this one reference person that whenever you have any question they are ready to help"* (FG2-P13). In fact, two participants indicated that because FRRP is mandatory, they enrolled in the program. While others, thought they would rather have it "*optional*" (FG1-P01).

## Discussion

### Summary

The objective of this study was to explore the views of FRRP participants about the training program and how to enhance it, however, our ultimate intent was also to provide a prescriptive approach for similar programs that the global community can learn from.

In this study, we found that residents' main interest is to become skilled researchers. The FRRP program in its current status had important offerings in terms of coordination, mentorship and capacity building, but with substantial limitations that need to be addressed to strengthen it. In terms of expectations, the residents perceived themselves as novice researchers and required support with basic skills. The most common reported difficulty was the identification of important and interesting research questions. In terms of the program coordination, participants appreciated having a timeline and deadlines, the expectation of a close follow-up by the mentor, the close follow-up by the program coordinator, and the use of the portal. However, some complained about the tight timeline. In terms of mentorship, participants appreciated the flexibility in choosing the mentor and their own research question. However, they complained about having few faculty members who conducted research and about the lack of commitment of some mentors. In terms of capacity building, the feedback was overall positive, and the participants pointed out that the FRRP program is very supportive and resourceful. Participants also reported some challenges (e.g., the capacity building sessions were not interactive enough). Participants made practical recommendations to address these different challenges.

### Strengths and limitations of the study

The main strength of this study is the use of several approaches to increase the rigor of our study, including training of interviewers, peer observation, debriefing, and duplicate data analysis. Also, this is the first study we are aware of to assess residents' views about training in medical research.

One limitation of the study is that we were not successful in recruiting participants from all the departments at AUBMC. However, we suspect that the views of those who participated are representative of our target population.

### Comparison to similar studies

We were not able to identify qualitative assessments of programs like ours. However, we did identify studies describing similar programs [9,10]. Kanna et al. assessed a dedicated "research rotation" at a university hospital in New York city to teach residents how to design and implement a successful research project [9]. They reported increased participation of residents in scholarly activities and their perception of research as being essential for practice of medicine. In another study, Hurst et al. described the research training of pediatric residents and clinical fellows as part of the Duke Pediatric Research Scholars Program for Physician-Scientist Development (DPRS) [10]. Participants were able to secure internal (department and university-

wide) and external funding and publish articles in peer-reviewed journals. Dagher et al. reported on a similar program conducted at the American University of Beirut in Lebanon but amongst undergraduate students [1]. Under this program, students volunteer to work with faculty members on clinical and basic science research, to learn about research firsthand. The authors reported that participants have co-authored publications in peer-reviewed journals with their respective advisors and rated the program positively.

### Implications for practice

Our findings emphasize the importance of understanding and trying to meet the expectations of residents when establishing research training programs. Indeed, we have built on these findings to introduce several changes to our program. Example of changes include modifying the timeline and deadlines of the program (Fig 1) and enhancing the program portal. A major implication of our findings is ensuring availability and commitment of faculty mentors and when needed, offer them training in mentorship skills. As a result, the program has encouraged coordination at the departmental level particularly for ensuring an optimal resident-to-mentor matching. Another implication is importance of providing capacity building activities to the program participants. In our case, we have revised the program educational offerings to better cover the ABCs of research using interactive and hands on sessions. It is important to conceive such programs as a major investment for participants and departments alike, aiming to enhance research productivity [11]. Medical educators interested in starting or improving similar programs can hopefully build on our findings, particularly the elements described in Table 1.

### Implications for future research

It would be important to have a follow up study, with both participants and advisors, to assess how the changes introduced as a result of this study have changed the research culture and the research productivity during and following training. More generally, research is needed to explore optimal ways to improve the research experience of clinical trainees.

### Conclusion

In order to provide residents and clinical fellows with a supervised and structured research experience, we have built on our findings to introduce several changes to our program such as ensuring the availability and commitment of faculty mentors, providing capacity building activities to the program participants and revising the program educational offerings to better cover the ABCs of research using mostly interactive and hands on sessions.

### Supporting information

**S1 Table. Participant comments pertaining to expectations of residents, program coordination, mentorship, and capacity building.**
(DOCX)

**S1 Appendix. Interview guide.**
(DOCX)

### Acknowledgments

We acknowledge the contribution of the volunteer Ms. Dina El Achi under the Medical Research Volunteer Program (MRVP) at the American University of Beirut, Dr. Zeina Akiki

and Ms. Rola El Rassi for their contribution to proposal writing, and Dr. Ibrahim El Mikati for his contribution to designing a figure.

## Author Contributions

**Conceptualization:** Lara Al-Hakim, Hani Tamim, Elie A. Akl.

**Data curation:** Pamela Abi Khalil, Dina El Achi, Lara Al-Hakim, Elie A. Akl.

**Formal analysis:** Pamela Abi Khalil, Gladys Honein-Abou Haidar, Dina El Achi, Elie A. Akl.

**Methodology:** Pamela Abi Khalil, Gladys Honein-Abou Haidar, Lara Al-Hakim, Elie A. Akl.

**Project administration:** Pamela Abi Khalil, Elie A. Akl.

**Supervision:** Pamela Abi Khalil, Elie A. Akl.

**Validation:** Pamela Abi Khalil.

**Writing – original draft:** Pamela Abi Khalil, Gladys Honein-Abou Haidar, Elie A. Akl.

**Writing – review & editing:** Pamela Abi Khalil, Gladys Honein-Abou Haidar, Dina El Achi, Lara Al-Hakim, Hani Tamim, Elie A. Akl.

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
