## [Decision Letter · Decision Letter 0]

19 Oct 2021

PONE-D-21-29406Views of medical residents on a research training program: a qualitative studyPLOS ONE

Dear Dr. Aki

Thank you for submitting your manuscript to PLOS ONE. After careful consideration, we feel that it has merit but does not fully meet PLOS ONE’s publication criteria as it currently stands. Therefore, we invite you to submit a revised version of the manuscript that addresses the points raised during the review process.

We look forward to receiving your revised manuscript.

Kind regards,

Mohamed A Yassin, MD

Academic Editor

PLOS ONE

Journal Requirements:

[-Dr. Elie Akl directs the Clinical Research Institute, under which the program that was evaluated operates.

-Dr. Hani Tamim directs the Fellowship and Residency Research Program, the program that was evaluated.

-Ms. Abi Khalil coordinates the Fellowship and Residency Research Program, the program that was evaluated.

-All remaining authors have nothing to disclose.] 

3. Please include your tables as part of your main manuscript and remove the individual files. Please note that supplementary tables (should remain/ be uploaded) as separate "supporting information" files

Reviewers' comments:

Reviewer's Responses to Questions

**Comments to the Author**

1. Is the manuscript technically sound, and do the data support the conclusions?

Reviewer #1: Yes

Reviewer #2: Yes

2. Has the statistical analysis been performed appropriately and rigorously? 

Reviewer #1: N/A

Reviewer #2: Yes

3. Have the authors made all data underlying the findings in their manuscript fully available?

Reviewer #1: Yes

Reviewer #2: Yes

4. Is the manuscript presented in an intelligible fashion and written in standard English?

Reviewer #1: Yes

Reviewer #2: Yes

5. Review Comments to the Author

Reviewer #1: The methodology of this article is sound however the intent and usefulness of this articles for other educators is not clear.

This article would be significantly strengthened if it was written with outward intent as opposed to how it is currently, which seems more of an internal evaluation of your program, as opposed to useful information others can use to start or improve their own programs. While you have included quotes in the supplementary table, it would be helpful to comment on recurring ideas you have observed in the main body which in the current format, is not clear and seems to be more of individual quotes brought forward by participants. Main emerging themes are not obvious to the reader.

Reviewer #2: Review:

Comments to the Author

Thank you for the opportunity to review " Views of medical residents on a research training program: a qualitative study." This a very interesting topic and is of remarkable importance and clinical relevance. Overall, this a well-structured and concise manuscript. The introduction is short and clear. The methodology is detailed and well explained. The results and discussion are well presented and clarified. Specific comments follow.

Title

1. Please check capitalization.

Methodology:

1. Page 6, the first paragraph under “overall design and research team – the font size is different than the other paragraphs, please adjust it.

2. Page 7. Sentence 1 under “data collection” section - What is meant by “month to month”.

6. PLOS authors have the option to publish the peer review history of their article (what does this mean?). If published, this will include your full peer review and any attached files.

Reviewer #1: No

Reviewer #2: No

---

## [Author Response · Author response to Decision Letter 0]

2 Dec 2021

Response: We have revised the manuscript to ensure it meets PLOS ONE's style requirements

[-Dr. Elie Akl directs the Clinical Research Institute, under which the program that was evaluated operates.

-Dr. Hani Tamim directs the Fellowship and Residency Research Program, the program that was evaluated.

-Ms. Abi Khalil coordinates the Fellowship and Residency Research Program, the program that was evaluated.

-All remaining authors have nothing to disclose.] 

Response: Thank you. We have included the above statement in the manuscript following the Competing Interests section.

Response: We have included the updated Competing Interests statement in our cover letter.

3. Please include your tables as part of your main manuscript and remove the individual files. Please note that supplementary tables (should remain/ be uploaded) as separate "supporting information" files

Response: The one table we had included in the first version was a very large one that we opted to include as a “supporting information” file. We have uploaded it separately. We might have created confusion by previously naming it table 1. We have now included another table in the main manuscript (please see below).

Reviewer #1

The methodology of this article is sound however the intent and usefulness of this articles for other educators is not clear. This article would be significantly strengthened if it was written with outward intent as opposed to how it is currently, which seems more of an internal evaluation of your program, as opposed to useful information others can use to start or improve their own programs. While you have included quotes in the supplementary table, it would be helpful to comment on recurring ideas you have observed in the main body which in the current format, is not clear and seems to be more of individual quotes brought forward by participants. Main emerging themes are not obvious to the reader.

Response: thank you for this important suggestion. We have addressed it by highlighting the themes that have emerged (expectations of residents, coordination, mentorship, and capacity building). We describe the latter three as the ‘pillars’ for a research training for residents, with dedicated subheadings in the results section. We have also added a table (table 1) that describes the elements pertinent to each of these three pillars. Medical educators interested in starting or improving similar programs would hopefully find this information to be useful. We have also replaced figure 2 with another figure that better represents the 3 pillars.

Reviewer #2

Thank you for the opportunity to review " Views of medical residents on a research training program: a qualitative study." This a very interesting topic and is of remarkable importance and clinical relevance. Overall, this a well-structured and concise manuscript. The introduction is short and clear. The methodology is detailed and well explained. The results and discussion are well presented and clarified.

Response: Thank you for the very positive feedback.

Title: Please check capitalization.

Response: The title has been modified 

Methodology: Page 6, the first paragraph under “overall design and research team – the font size is different than the other paragraphs, please adjust it.

Response: Font size has been adjusted 

Page 7. Sentence 1 under “data collection” section - What is meant by “month to month”.

Response: Apologies for the confusion. We have now specified the exact months when we conducted the data collection)

---

## [Editor Report · Decision Letter 1]

6 Dec 2021

Views of medical residents on a research training program: a qualitative study

PONE-D-21-29406R1

Dear Dr.Aki,

We’re pleased to inform you that your manuscript has been judged scientifically suitable for publication and will be formally accepted for publication once it meets all outstanding technical requirements.

Kind regards,

Mohamed A Yassin, MD

Academic Editor

PLOS ONE

Additional Editor Comments (optional):

Since the required modifications by reviewers were addressed , the manuscript can be accepted in its current form
---

## [Editor Report · Acceptance letter]

23 Dec 2021

PONE-D-21-29406R1 

Views of medical residents on a research training program: a qualitative study 

Dear Dr. Akl:

I'm pleased to inform you that your manuscript has been deemed suitable for publication in PLOS ONE. Congratulations! Your manuscript is now with our production department. 

Kind regards, 

on behalf of

Dr. Mohamed A Yassin 

Academic Editor

PLOS ONE